# Maternal Age, the Disparity across Regions and Their Correlation to Sudden Infant Death Syndrome in Taiwan: A Nationwide Cohort Study

**DOI:** 10.3390/children8090771

**Published:** 2021-09-01

**Authors:** Lin-Yi Huang, Wan-Ju Chen, Yung-Ning Yang, Chien-Yi Wu, Pei-Ling Wu, Shu-Leei Tey, San-Nan Yang, Hsien-Kuan Liu

**Affiliations:** 1Department of Pediatrics, E-DA Hospital, Kaohsiung 82445, Taiwan; pedptch05680@gmail.com (L.-Y.H.); ru74225@hotmail.com (W.-J.C.); ancaly@yahoo.com.tw (Y.-N.Y.); wucyi1228@yahoo.com.tw (C.-Y.W.); peiling0420@gmail.com (P.-L.W.); djsr2000@hotmail.com (S.-L.T.); y520729@gmail.com (S.-N.Y.); 2College of Medicine, I-Shou University, Kaohsiung 82445, Taiwan

**Keywords:** foreign immigrant, foreign spouse, maternal age, region inequality, sudden infant death syndrome, teenage pregnancy

## Abstract

Sudden infant death syndrome (SIDS) has always been a regrettable issue for families. After sleeping in the supine position was proposed, the incidence of SIDS declined dramatically worldwide. However, SIDS still accounts for the top 10 causes of infant deaths in Taiwan. Recognizing the risk factors and attempting to minimize these cases are imperative. We obtained information on cases with SIDS from the National Health Insurance Research Database in Taiwan and interconnected it with the Taiwan Maternal and Child Health Database to acquire infant–maternal basal characteristics between 2004 and 2017. The SIDS subjects were matched 1:10 considering gestational age to normal infants. After case selection, a total of 953 SIDS cases were included. Compared with healthy infants, SIDS infants had younger parents, lower birth weight, and lower Apgar scores. After adjusting for potential confounders, infants with mothers aged <20 years had 2.81 times higher risk of SIDS. Moreover, infants in the non-eastern region had a significantly lower risk of SIDS than those in the eastern region. We concluded that infants of young mothers (especially maternal age <20 years) and infants in the eastern region of Taiwan had a higher risk of SIDS than their counterparts.

## 1. Introduction

Sudden infant death syndrome (SIDS) is defined as the unexpected death of babies younger than 12 months of age without any adequate cause of death even after a thorough examination and clinical history review [1,2,3]. To date, many studies have identified the relevant risk factors associated with SIDS. Intrinsic factors (i.e., genetics, smoke exposure, prematurity, restricted intrauterine growth), extrinsic factors (i.e., prone or side sleeping position, bed-sharing), ambient heat, and air pollution were all proposed to be relevant risks factors for SIDS [1,4,5,6,7,8]. Among these factors, sleeping in the prone or side-lying position is the most recognized factor that leads to SIDS [1,9]. Since the recommendation of the sleeping position policy in 1992 by the American Academy of Pediatrics, the incidence of SIDS has declined dramatically [10]. While there are many medical factors correlated with SIDS, some public health factors have also been considered. Among the public health issues, pregnancy at an early age and regional inequality of development and medical resources are essential factors that need to be investigated in Taiwan.

Young maternal age has been shown to increase the risk of SIDS, according to previous studies [11,12]. Despite the decrease in the incidence of childbirth by adolescents and the advancement of health education as a result of socioeconomic progress, there was still a portion of young-age pregnancies in Taiwan [13]. Teenage pregnancy has been consistently associated with a higher incidence of SIDS. This was due to illegal substance abuse, mental disorders, socioeconomic disadvantage, high incidence of developing anemia during pregnancy, preterm delivery, and delivery of low birth-weight babies [14,15,16]. Therefore, the relationship between SIDS and teenage pregnancy in Taiwan should be evaluated.

Furthermore, the distribution of medical resources remains unequal in Taiwan. The number of medical centers differed between urban cities and rural regions, especially shortage in the eastern area. The shortage of medical staff in the outer areas of Taiwan is also a vital issue [17,18]. There have also been reports that the infant mortality rate is higher in rural areas than in urban areas [19,20,21]. As explained earlier, since medical resources and accessibility were unequal in Taiwan, the incidence of SIDS in different areas is worth exploring.

Due to the customs and culture, parents in Taiwan are usually unwilling to perform post-mortem examination. The diagnosis of SIDS mainly made by pediatricians’ judgement after thoroughly history taking and examinations among all regions in Taiwan. Furthermore, since the numbers of SIDS cases are relatively small regardless of the hospital, gathering all SIDS cases for advanced analysis might help us understanding the situation in Taiwan. To bridge these gaps due to lack of sufficient reports on the issue, we designed a study to investigate the relationship between SIDS, maternal age, and geographic distribution. This study aimed to determine the incidence of SIDS and other associated factors that could further help improving public health in Taiwan.

## 2. Materials and Methods

### 2.1. Data Source

The healthcare system in Taiwan, which is known as National Health Insurance (NHI), implemented in 1995, is a compulsory social insurance program. This system has the advantage of good accessibility and consists of a total population of over 99%. The National Health Insurance Research Database (NHIRD), derived from NHI, can be analyzed for records of public health research [22,23,24]. The death registration database (which was initiated in 1997) was accessed from the NHIRD, and the age of death, cause of death (using the definition from the International Classification of Diseases [ICD]), and place of death were recorded [25].

The Birth Certificate Application (BCA), which was developed by the Ministry of Health and Welfare in Taiwan, contains information about birth conditions. Medical facilities have to register birth information of all newborns within 7 days of birth [26]. The registered data in BCA contained the parents’ and newborns’ basal characteristics, such as birth weight, gestational age, sex, mode of delivery, and age of parents. However, since each individual’s death is shown in the NHIRD while the birth characteristics are registered in the BCA, gathering these data for analysis remains challenging.

To investigate the correlation between newborns who died and their birth conditions and basal characteristics, the Taiwan Maternal and Child Health Database (TMCHD), which was authorized by the Ministry of Health and Welfare, was set up to integrate these data from 2004 [27]. Hence, using the TMCHD system, we could connect and acquire information about the offspring and their parents for further analysis. The individual data in the TMCHD were all de-identified, and the requirement for informed consent was waived. We obtained approval from the Institutional Review Board of E-Da Hospital (EMRP-108-061).

### 2.2. Study Population

Infants younger than 1 year who had sudden deaths between 2004 and 2017 and whose diagnostic codes were recorded as ICD-9-CM 798.0 and ICD-10-CM R95 from the NHIRD were selected as SIDS subjects. The date of birth was used as the index day in this cohort. All subjects were observed until death or 1 year after the index day. The identities of the non-SIDS subjects were concealed for at least 1 year during the follow-up period. Cases of twins or multiple births, congenital abnormalities, stillbirth, deaths during the first day after birth, and patients aged ≥1 year before death were excluded. The SIDS subjects were matched 1:10 to normal infants with respect to gestational age. Demographic data such as mother’s age, father’s age, sex of the infant, mother’s nationality, birth weight, mode of delivery, Apgar score (at 1 and 5 min), maternal and obstetric risk factors (anemia and hypertensive disorder, premature rupture of membranes, and fetal distress) were collected for further adjustment. The two cohorts were pooled to explore the association of maternal age at pregnancy on sudden infant death. The mothers’ age at pregnancy was used as a standard to divide the subjects into two groups of “maternal age < 20” and “maternal age ≥ 20” following the definition of teenage pregnancy or adolescent pregnancy [28]. Furthermore, we divided Taiwan into separate areas for analysis according to the geographical location and regional inequality of development and medical resources [29]. This method helped to analyze the basic data of the two groups. Sudden deaths in the two groups were also selected for in-depth analysis.

### 2.3. Survival Data

The data from the national death registry maintained by the Ministry of Health and Welfare of Taiwan for cases caused by related diseases were also considered. They were used to confirm the survival of each case in the two cohort populations in this study.

### 2.4. Statistical Analysis

All data management and hazard ratio (HR) calculations were performed using the Statistical Analysis System (SAS) software for Windows (version 9.4; SAS Institute, Cary, NC, USA). Proportions were used to represent categorical variables and means ± standard deviations were used for continuous variables. The unpaired Student’s *t*-test (for continuous variables) and the chi-square test (for categorical variables) were used to compare variables between the two groups. The proportion of survival of newborns between the mothers aged< or ≥20 years was assessed using the Kaplan–Meier analysis, and the significance was calculated using the log-rank test. All *p* values less than 0.05 were considered significant.

## 3. Results

Between 2004 and 2017, a total of 1915 infants diagnosed with SIDS were identified from the death registration database of the NHIRD. Since SIDS was defined as death before 1 year of age, infants born before 2004 were excluded (*n* = 725). The deaths that occurred during the perinatal period but coded with 798.0 (ICD 9) and R95 (ICD 10) (*n* = 2) were also excluded. Cases of twins or multiple births (*n* = 101), congenital abnormalities (*n* = 0), and patients aged ≥1 year before death (*n* = 134) were excluded. After exclusion, 953 patients with SIDS were enrolled. Furthermore, 9530 healthy subjects were randomly recruited for comparison with the 953 SIDS infants (Figure 1). The total incidence of SIDS declined gradually during the study period (0.54‰ in 2004 and 0.109‰ in 2017) (Figure 2). Since prematurity is a significant factor that correlates with SIDS [30], we designed a model with the same gestational age to eliminate the effect of SIDS. We also designed the model with 10 times more healthy infants than SIDS infants and compared them as shown in Table 1. Among the SIDS group, the birth weight was less (2842.78 ± 638.06 and 2913.5 ± 658.04 g for SIDS and healthy infants, respectively; *p* < 0.05). A higher proportion of mothers were native citizens (Taiwanese) in the SIDS group, and the age of the parents was less than that of those of healthy infants. The SIDS group also showed lower Apgar scores at 1 min and 5 min (all *p* < 0.001) and a higher incidence of fetal distress during the perinatal period (*p* < 0.05). The incidence of SIDS also showed difference between regions in Taiwan (*p* < 0.05). The data showed no significant difference in other maternal and obstetric risk factors, such as maternal anemia and hypertensive disorders. In contrast, the incidence of premature rupture of membranes was higher in the healthy infant cohort (*p* < 0.05).

### 3.1. Prediction for the Occurrence of SIDS

After adjusting for confounding factors to predict the occurrence of SIDS, offspring of mothers younger than 20 years had a 2.81 times higher risk of sudden death than those of mothers older than 20 years. Analyzing the difference across regions, infants born in the non-eastern region rather than in the eastern region of Taiwan showed a significantly lower SIDS risk (HR: 0.42, 0.42, 0.37, 0.37, and 0.35 in the Northern, North Central, Central, South Central, and Southern areas, respectively; all *p* < 0.05). The incidence of post-natal mortality before 1 year old was shown in Appendix A. Higher mortality rate before 1 year old in the Eastern area was also noted. A higher Apgar score at 1 min was associated with a lower risk of SIDS (HR: 0.73, *p* < 0.05). Furthermore, younger age of fathers and lower gestational weeks were also significantly associated with SIDS (Table 2).

### 3.2. Comparison of the Differences between Two Groups using 20 Years of Maternal Age as the Reference in the Total Study Population

The total study population was divided into two groups following the definition of teenage pregnancy (maternal age < 20 years vs. maternal age ≥ 20 years). Lower birth weight (2679.94 ± 734.43 vs. 2913.01 ± 653.35 g), younger paternal age (25.19 ± 5.26 vs. 33.88 ± 5.45 years), less gestational age of neonates (36.56 ± 4.43 vs. 37.36 ± 3.27 weeks), lower Apgar score at 1 min (7.57 ± 1.92 vs. 8.04 ± 1.39) and 5 min (9.12 ± 2.86 vs. 9.79 ± 1.42), and more cases of SIDS (31.84% vs. 8.5%) were observed in the maternal age <20 years group than in the maternal age ≥20 years group (*p* < 0.001 for all the above parameters). Furthermore, the proportion of maternal anemia in the group aged <20 years was significantly higher than that in the group aged ≥20 years (Table 3). In contrast, there were no differences among maternal hypertensive disorder, the proportion of premature rupture of membranes, and fetal distress episodes during the perinatal period between the two groups (Table 3).

### 3.3. Proportion of Survival between the Maternal Age ≥20 and <20 Years Cohorts

The Kaplan–Meier survival curve was drawn to assess survival rate until 12 months of age according to the definition of SIDS. The curve is presented in Figure 3. The 1-year survival rate was lower in neonates of mothers aged <20 years (log-rank test, *p* < 0.0001).

## 4. Discussion

The infant mortality rate (IMR), defined as the number of deaths of children under 1 year of age, is regarded as a basic indicator of public health worldwide [31]. IMR has gradually declined over the past decades worldwide, including Taiwan. This is owing to advancements in medical technologies, promotion of public health education, and increase in medical manpower and facilities [32,33]. Even though Taiwan was not included in the analysis of the Organization for Economic Cooperation and Development (OECD) countries, IMR still ranked moderately higher in Taiwan than in other European countries [34]. In our study, the SIDS incidence was observed to decrease from 5.41 in 10,000 children to 1.09 in 10,000 children. However, even if the rate of SIDS gradually declined, it still accounted for the top 10 causes of infant deaths [35]. Meanwhile, the trend of Taiwan’s fertility rate is decreasing [36]. Under these circumstances, measures should be implemented to minimize SIDS cases. 

The proportion of teen pregnancies in Taiwan remains higher than that in some European countries [37]. A local study revealed that this might be related to poor contraceptive knowledge and low socioeconomic status [38]. Moreover, teen child-bearing women are more likely to have lower educational growth, more adverse pregnancy complications, and even more postpartum depression [39,40]. In our study, infants of mothers aged <20 years had 2.81 times higher SIDS risk than those of mothers aged ≥20 years. In addition, our results also suggested that infants with mothers aged <20 years tend to have a small gestational age, which is also a key factor associated with SIDS [1,2]. Similar to maternal age, younger paternal age was also significantly associated with higher SIDS occurrence according to our results. This might also be another social issue, which is that of young-age marriage, that we should pay attention to. These results imply that teenage pregnancies should be taken seriously. Comprehensive sex education, not only for young women but also for young men, should be promoted by the government to effectively prevent and decrease the incidence of teenage pregnancies [41,42] and, thus, decrease the incidence of SIDS.

Our results also indicated that there are notable geographic variations in the SIDS occurrence rate. The eastern region had a higher incidence of SIDS than the non-eastern region. This may be owing to the lower medical personnel density in the eastern region than in other regions [29]. In Taiwan, the availability, accessibility, affordability, and acceptability of medical resources were remarkable after the NHI was implemented in 1995. However, insufficient medical resources in the eastern region of Taiwan remain a significant issue [29]. Previous studies have demonstrated a disparity in SIDS incidence between urban and rural areas and in income levels. SIDS incidence was higher in rural areas and the impoverished population [21,43]. This could explain the higher SIDS occurrence in the eastern region of Taiwan. Narrowing the urban–rural and inter-regional gaps in development will play an important role in a child’s health environment. The other possible reason that might lead to regional disparity is the different diagnostic approach of SIDS. In Taiwan, the public health insurance was implemented for a long time, and all the diagnostic approach was almost consistent. Furthermore, regular conferences were held to discuss about SIDS and child health care issues so that the diagnostic process goes identical. Owing to above reason, the different incidence of SIDS might not be associated with the different diagnostic approach.

Moreover, previous study also revealed that the child maltreatment rate in the eastern region ranked first in Taiwan [44]. Since SIDS is a diagnosis after excluding all explainable causes, it is important to distinguish it from child maltreatment. However, SIDS is difficult to distinguish completely from fatal child abuse [45]. In Taiwan, most SIDS cases were diagnosed clinically without autopsy. However, even under autopsy, SIDS is difficult to distinguish from suffocation caused by soft objects, either accidentally or deliberately [45]. Thus, the more SIDS cases that have been reported in the eastern region might be because they were caused by child maltreatment rather than SIDS. Regardless, it remains a huge burden on society and an issue that needs to be resolved urgently.

Another result worth noting is the incidence of SIDS between native Taiwanese and immigrated mothers. The relationship between maternal mental health and offspring health has always been a topic of investigation. Previous studies showed that SIDS is tightly correlated with maternal psychiatric disorders, especially perinatal depression [46,47]. In Taiwan, there has been a large increase in transnational marriages from China and Southeast Asia (Vietnam and Indonesia) [48]. Many of these foreign spouses face some psychiatric disorders, especially those who cannot speak the local language, according to a previous study [49,50]. However, according to our results, there was no obvious difference in SIDS between native and foreign-born infants. This could be explained by the social and cultural background of foreign spouses in Taiwan. Most foreign mothers migrated from China [48]. In fact, there is no apparent discrepancy in the background culture, language, and religion between China and Taiwan. Therefore, immigrated women from China could easily merge into society without substantial effort, thus reasonably explaining our results.

This study had some limitations. First, it was a retrospective study, and data were obtained from the NHIRD based on ICD-9-CM and ICD-10-CM codes. Misclassification of diseases may occur. We could not exclude selection bias because of the retrospective nature of the study. Second, the diagnosis of SIDS in Taiwan is based on pediatricians’ clinical judgement after thoroughly history taking and excluding all the possible causes. Post-mortem examination in Taiwan is not routinely arranged unless the pediatrician could not make a diagnosis. In previous literature (written in Chinese), the post-mortem examination rate was only 6.03% between 1996 to 2005 [51]. However, there is a certain consensus on the diagnosis of SIDS even in various regions of Taiwan. Owing to this reason, the practices in different regions of Taiwan are roughly the same. Third, it is difficult to make a precise clinical diagnosis of SIDS. SIDS is diagnosed after excluding all reasonable causes of death. However, some hidden issues may not be found immediately while the death occurred, for example, child maltreatment. However, since the child protection medical service demonstration center was established in Taiwan in August 2014, the ability to recognize child maltreatment by medical staff has enhanced and more attention has been paid to child maltreatment [52]. Thus, the misjudgment of SIDS cases and child maltreatment has decreased. Fourth, infant death data were obtained from the NHI database. However, the maternal data of the cases were difficult to thoroughly link. Though we integrated these data with TMCHD, data on some maternal risk factors that would affect SIDS (i.e., gestational diabetes, smoking) could not be acquired completely. 

## 5. Conclusions

In conclusion, the trend of SIDS in Taiwan gradually declined annually after more public health promotion and attached great importance to children’s safety. The risk factors for SIDS analyzed using the national population data revealed that maternal age and regional distribution were the most significant factors. Offspring of younger mothers, especially of aged <20 years, had a higher incidence of SIDS. Infants in the non-eastern region of Taiwan, which is an area with relatively abundant medical resources and is more developed than the eastern area, had a lower risk of SIDS. Moreover, there was no discrepancy in SIDS incidence between the offspring of foreign immigrants and native mothers. Since there was no significant difference between the immigrant and native mothers, we might consider dividing the foreign spouses into China and Southeast Asia to see if there will be different results in the further study.

## Figures and Tables

**Figure 1 children-08-00771-f001:**
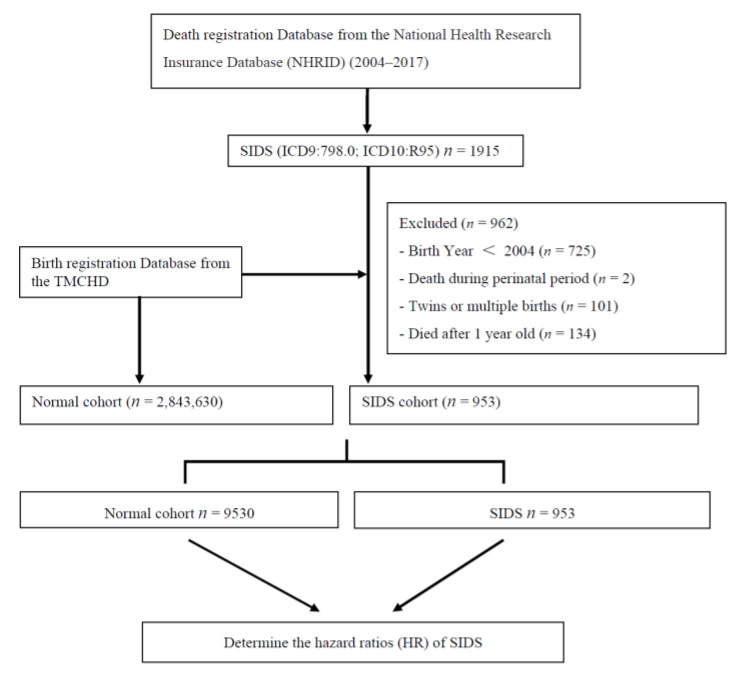
Study flowchart.

**Figure 2 children-08-00771-f002:**
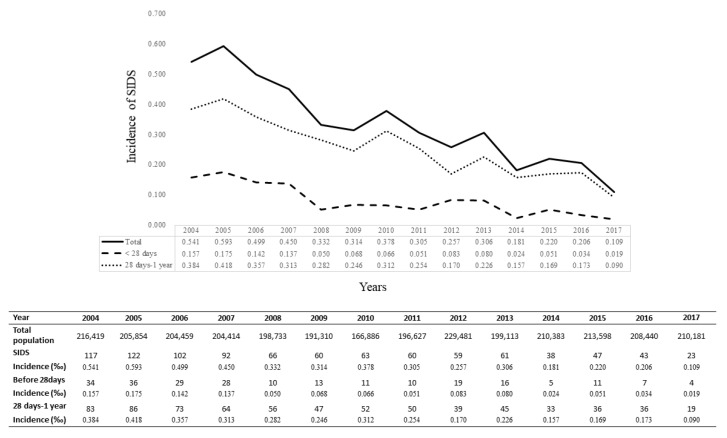
Sudden infant death syndrome (SIDS) population and the incidence of SIDS in Taiwan between 2004 to 2017.

**Figure 3 children-08-00771-f003:**
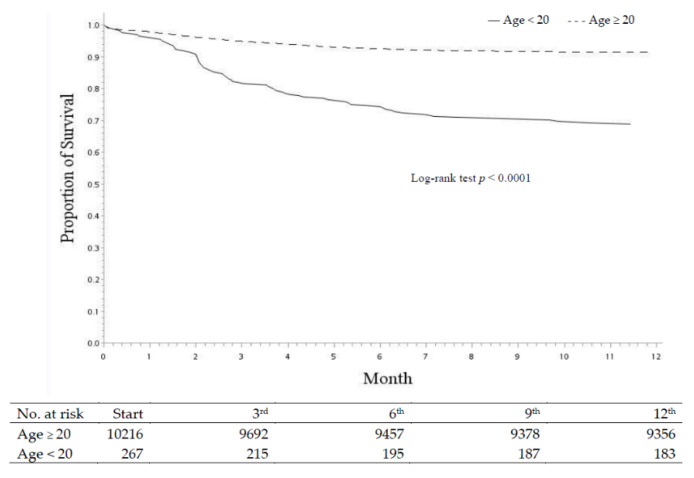
Proportion of survival of offspring between the maternal age ≥ 20 and < 20 years cohorts.

**Table 1 children-08-00771-t001:** Baseline characteristics of the study population coded with SIDS, Taiwan, 2004–2017.

	Normal Cohort *n* = 9530	SIDS Cohort *n* = 953	*p*-Value
Gestational age	37.34 ± 3.31	37.34 ± 3.31	>0.9999
Infant sex			0.0013
Boy	4911 (51.53%)	543 (56.98%)	
Girl	4619 (48.47%)	410 (43.02%)	
Birth weight	2913.5 ± 658.04	2842.78 ± 638.06	0.0015
Native citizen (mother)	8744 (91.75%)	913 (95.8%)	<0.0001
Age of father	33.84 ± 5.49	32.8 ± 6.10	<0.0001
Age of mother	30.73 ± 5.03	28.18 ± 5.87	<0.0001
Region (in Taiwan)			0.0039
Northern	2869 (30.1%)	249 (26.13%)	
North Central	1761 (18.48%)	213 (22.35%)	
Central	1908 (20.02%)	181 (18.99%)	
South Central	1238 (12.99%)	123 (12.91%)	
Southern	1409 (14.78%)	138 (14.48%)	
Eastern	345 (3.62%)	49 (5.14%)	
Mode of delivery			0.0056
Vagina	5813 (61%)	625 (65.58%)	
Cesarean section	3717 (39%)	328 (34.42%)	
Apgar score (1 min)	8.07 ± 1.34	7.66 ± 1.96	<0.0001
Apgar score (5 min)	9.84 ± 1.27	9.2 ± 2.72	<0.0001
Maternal and obstetric risk factors			
Anemia	79 (0.83%)	6 (0.63%)	0.5129
Hypertensive disorder	186 (1.95%)	15 (1.57%)	0.4175
Premature rupture of membranes	285 (2.99%)	10 (1.05%)	0.0006
Fetal distress	142 (1.49%)	28 (2.94%)	0.0007

SIDS: Sudden infant death syndrome.

**Table 2 children-08-00771-t002:** Predictions for the occurrence of SIDS using multiple regression.

	Crude HR (95% CI)	*p*	Adjusted HR (95% CI)	*p*
Maternal age (<20 vs. ≥20 years)	4.22 (3.38–5.27)	<0.0001	2.81 (1.43–5.50)	<0.01
Gender	1.23 (1.08–1.40)	<0.01	1.20 (0.93–1.56)	0.166
Birth weight	1.00 (1.00–1.00)	<0.01	1.00 (1.00–1.00)	0.083
Paternal age	0.97 (0.95–0.98)	<0.0001	0.96 (0.94–0.99)	<0.01
Native citizen (mother)	1.99 (1.45–2.74)	<0.0001	1.78 (0.90–3.50)	0.095
Region (in Taiwan)				
Northern	0.62 (0.46–0.84)	<0.01	0.42 (0.20–0.88)	<0.05
North Central	0.86 (0.63–1.17)	0.326	0.42 (0.20–0.89)	<0.05
Central	0.68 (0.50–0.93)	<0.05	0.37 (0.18–0.80)	<0.05
South Central	0.71 (0.51–0.99)	<0.05	0.37 (0.17–0.82)	<0.05
Southern	0.70 (0.51–0.97)	<0.05	0.35 (0.16–0.77)	<0.01
Eastern	REF.		REF.	
Gestational age	1.00 (0.98–1.02)	0.868	1.11 (1.01–1.21)	<0.05
Mode of delivery	1.21 (1.06–1.38)	<0.01	1.18 (0.88–1.58)	0.262
Apgar score (1 min)	0.85 (0.82–0.88)	<0.0001	0.73 (0.55–0.96)	<0.05
Apgar score (5 min)	0.85 (0.81–0.88)	<0.0001	0.97 (0.76–1.24)	0.832
Maternal and obstetric risk factors				
Anemia	0.76 (0.34–1.70)	0.505	1.12 (0.16–8.11)	0.910
Hypertensive disorder	0.82 (0.49–1.36)	0.434	1.35 (0.49–3.70)	0.557
Premature rupture of membranes	0.36 (0.19–0.66)	<0.05	0.43 (0.13–1.36)	0.150
Fetal distress	1.98 (1.36–2.88)	<0.05	0.61 (0.14–2.58)	0.502

SIDS, Sudden infant death syndrome; HR, Hazard Ratio; CI, Confidence interval. REF, reference.

**Table 3 children-08-00771-t003:** Basal characteristics between the two groups divided by maternal age (<20 and ≥20 years).

	Age < 20 Years Cohort (*n* = 267)	Age ≥ 20 Years Cohort (*n* = 10,216)	*p*
Cases of SIDS	85 (31.84%)	868 (8.50%)	<0.0001
Birth weight	2679.94 ± 734.43	2913.01 ± 653.35	<0.0001
Native citizen (mother)	252 (94.38%)	9405 (92.06%)	0.1647
Paternal age	25.19 ± 5.26	33.88 ± 5.45	<0.0001
Maternal age	18.15 ± 1.31	30.82 ± 4.82	<0.0001
Region (in Taiwan)			<0.0001
Northern	47 (17.60%)	3071 (30.06%)	
North Central	59 (22.10%)	1915 (18.75%)	
Central	56 (20.97%)	2033 (19.90%)	
South Central	43 (16.10%)	1318 (12.90%)	
Southern	42 (15.73%)	1505 (14.73%)	
Eastern	20 (7.49%)	374 (3.66%)	
Gestational age	36.56 ± 4.43	37.36 ± 3.27	0.0001
Mode of delivery			<0.0001
Vagina	218 (81.65%)	6220 (60.88%)	
Cesarean section	49 (18.35%)	3996 (39.12%)	
Apgar score (1 min)	7.57 ± 1.92	8.04 ± 1.39	<0.0001
Apgar score (5 min)	9.12 ± 2.86	9.79 ± 1.42	0.0002
Maternal and obstetric risk factors			
Anemia	7 (2.62%)	78 (0.76%)	0.0008
Hypertensive disorder	3 (1.12%)	198 (1.94%)	0.3380
Premature rupture of membranes	6 (2.25%)	289 (2.83%)	0.5704
Fetal distress	3 (1.12%)	167 (1.63%)	0.5139

SIDS, Sudden infant death syndrome.

## Data Availability

The data presented in this study are available on request from the corresponding author. The data are not publicly available because they report private information about participants.

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
