# Peer review of "Maternal Age, the Disparity across Regions and Their Correlation to Sudden Infant Death Syndrome in Taiwan: A Nationwide Cohort Study"

_children, 2021, doi:10.3390/children8090771_

Round 1
Reviewer 1 Report
I am concerned that by limiting the analysis to SIDS only there is the risk of introducing significant bias.
It is well known that the rate of ICD10 R95 diagnosis of SIDS varies considerably between and within different countries, and many cases that are identical to SIDS are instead labelled as R98 (unascertained) or other causes (see Taylor BJ, Garstang J, Engelberts A, et al International comparison of sudden unexpected death in infancy rates using a newly proposed set of cause-of-death codes Archives of Disease in Childhood 2015;100:1018-1023.) In order to be certain of the paper’s conclusion, the authors need to consider the impact of potential SIDS cases being misclassified. This is a major weakness of the paper.
The authors compare absolute numbers of SIDS cases between the regions, without stating the population size and birth rate for the regions this comparison is not helpful.
I am not able to comment on the statistics in any detail and recommend a specialist statistics review.
More specific comments are as follows:
Introduction:
Please clarify how a SIDS diagnosis is reached in Taiwan. What proportion of cases had post-mortem examination with ancillary testing, a detailed medical history, death scene analysis and multi-professional case review? Is this same for all regions of the country?
Methods
Study population: It’s a little confusing using the terms ‘SIDS subjects’ and ‘all subjects’ ‘remaining subjects’. The non-SIDS subjects are controls.
What was the definition of perinatal period – first week or first month of life? Please clarify.
Results
Figure 2 – Is the incidence of SIDS per 1000 live born infants? It would be helpful to see total post-natal mortality for comparison too, as this will show that any reduction in SIDS rates is genuine and not a result of diagnostic shift.
Table 1. There are considerably less SIDS cases and controls from the Eastern region, and the percentage of cases and controls matches for each region. The methods state that controls were only matched on gestational age, but it looks as if they are matched by region too. This needs clarifying. The infant mortality rate for each region should be given to enable the reader to determine if the apparent difference in SIDS numbers between regions is meaningful.
Table 3. What is the figure in () in each column?
This may be my misunderstanding, but I don’t follow why the Eastern region has the highest rate of SIDS but the lowest number of cases.
Discussion.
The 3rd paragraph discusses the regional variation but without knowing the IMR and post-natal mortality rate for each region it is difficult to be certain. A major confounder could be that different diagnostic processes are used in different regions – so SIDS cases are not identified in all regions or over-diagnosed in the East. This needs further elaboration. Are there differences in infant sleep and care practices in the East compared to other regions which could explain the different SIDS rates? You suggest that medical resources prevents SIDS – this needs more explanation, as safe-sleep practices are not resource intensive and can be provided effectively by community educators.
The final paragraph talks on child maltreatment and SIDS being misdiagnosed. There needs to be more detail on the diagnostic processes for SIDS in Taiwan to understand this, are there any data on how many SIDS cases may actually be child abuse? I’m not clear how a child maltreatment clinic would help SIDS misdiagnosis unless SIDS cases are investigated by child maltreatment specialists – this links with my earlier comment that the introduction should include full details of how SIDS is diagnosed locally.
Conclusion.
The first sentence should be rewritten. SIDS in not prevented by access to medical services, but by community education, reduction in social deprivation etc.
Reviewer 2 Report
This is a remarkable study into the maternofetal and demographic risk factors for SIDS. The study obtained data from the National Health Insurance Research Database snd the Taiwan Maternal and Child Health Database and has been able to link the two to provide revealing and significant findings. The paper is straightforward and presented in a clear and uncomplicated manner. Perhaps, the only criticism is the failure to try and interpret and speculate on what could underlie their findings: mention should be made of pregnancy and birth factors. These indicate a host that is perhaps more vulnerable to environmental influences (such as infection). And similarly, postnatal factors point in the same direction.
